# Programmed Cell Death Protein-1 Upregulation in Response to SARS-CoV-2 in Juvenile Idiopathic Arthritis: A Case-Control Study

**DOI:** 10.3390/jcm11144060

**Published:** 2022-07-13

**Authors:** Violetta Opoka-Winiarska, Ewelina Grywalska, Izabela Korona-Głowniak, Izabela Morawska, Krzysztof Gosik, Anna Malm, Jacek Roliński

**Affiliations:** 1Department of Paediatric Pulmonology and Rheumatology, Medical University of Lublin, Gebali 6, 20-093 Lublin, Poland; 2Department of Experimental Immunology, Medical University of Lublin, Chodzki 4a, 20-093 Lublin, Poland; ewelina.grywalska@umlub.pl (E.G.); krzysztof.gosik@umlub.pl (K.G.); 3Department of Pharmaceutical Microbiology, Medical University of Lublin, Chodzki 1, 20-093 Lublin, Poland; iza.glowniak@umlub.pl (I.K.-G.); anna.malm@umlub.pl (A.M.); 4Department of Clinical Immunology and Immunotherapy, Medical University of Lublin, Chodzki 4a, 20-093 Lublin, Poland; izabelamorawska19@gmail.com (I.M.); jacek.rolinski@umlub.pl (J.R.)

**Keywords:** JIA, COVID-19, SARS-CoV-2, programmed cell death protein 1 (PD-1), DMARD

## Abstract

Currently, data regarding the impact of COVID-19 disease (caused by SARS-CoV-2) on patients with childhood rheumatic diseases are significantly limited. To assess the possible connection, we measured levels of IgA and IgG anti-SARS-CoV-2 antibodies in children with juvenile idiopathic arthritis (JIA) and a control group during the pandemic, prior to the introduction of anti-COVID-19 vaccination. We assessed levels of PD-1 suppressive molecule and inflammatory markers in patients and correlated those results with serological response to SARS-CoV-2. In JIA patients, the activity of the disease was assessed using the Juvenile Arthritis Disease Activity Score 71 (JADAS 71) scale. The study consisted of 96 children, 65 diagnosed with JIA, treated with antirheumatic drugs, and 31 healthy volunteers. In patients with JIA, significantly higher levels of SARS-CoV-2 antibodies in the IgA and IgG were demonstrated compared to the control group. We also found significantly higher serum PD-1 levels in JIA patients and control volunteers who were seropositive for SARS-CoV-2 IgA or IgG antibodies compared to those who were seronegative. The humoral immune response to SARS-CoV-2 infection is associated with the persistent upregulation of PD-1 expression in both JIA patients and healthy children. The clinical significance of the detected disorder requires further careful observation.

## 1. Introduction

The impact of SARS-CoV-2 infection on the immune system of patients with rheumatic disease remains unclear and requires explanation. Not only is the function of the immune system in rheumatic disease disturbed, with a predominance of overactivity with autoimmunity, but additionally SARS-CoV-2 infection can significantly change the function of the immune system, even in previously healthy people. Moreover, on this topic, the effects of anti-inflammatory and immunosuppressive drugs should be considered [1].

The immune defense in COVID-19 infection is balanced between the effects of suppression and an inadequate immune response with overactivity resulting in a systemic inflammatory process. An interesting study about the immune checkpoints with a proven role in balancing immunological response in COVID-19 has been published [2].

The course of COVID-19 in children with rheumatic diseases is carefully monitored [3]. Our recently published study showed no unequivocal correlation between the presence of SARS-CoV-2 antibodies and juvenile idiopathic arthritis (JIA) activity [4].

Programmed cell death-1 protein (PD-1 or CD279) is a member of the CD28 superfamily and is expressed on CD4+ and CD8+ T cells, NK cells, B cells, and activated monocytes. Binding to the ligands PD-L1 and PD-L2 on T cells, B cells, dendritic cells (DCs), and macrophages, PD-1 conducts its inhibitory function to regulate T cell activation, tolerance, and immune-mediated tissue damage. The importance of the PD-1 pathway has recently been investigated in autoimmune diseases in adults, but as only a few publications concern JIA, an understanding of its role in this condition is still significantly limited [5].

## 2. Results

### 2.1. Baseline Characteristics of the Patients

In total, 65 patients with JIA (aged 2–18 years), and 31 healthy children (aged 2–18 years) were included. All JIA patients were diagnosed according to the ILAR 2001 classification [5] prior to study enrollment and were treated for at least 6 weeks with at least one conventional or biological disease-modifying antirheumatic drug (DMARDs).

Baseline demographic and clinical characteristics of the studied groups are shown in Table 1.

### 2.2. Clinical Characteristics of JIA Patients Seropositive and Seronegative for Anti-SARS-CoV-2

A significantly higher anti-SARS-CoV-2 IgA and IgG ratio was demonstrated in JIA patients compared to the control group (Table 2). A total of 22 (33.88%) JIA patients had a positive result of at least of either anti-SARS-CoV-2 IgA or IgG antibodies, including 13 children (20%) with positive results in both classes.

In the control group, only four (12.9%) children tested positive for one of the antibody classes, while two (6.4%) of them had a positive result in both classes.

COVID-19 confirmed by SARS-CoV-2 RT-PCR test (4–8 weeks before our tests) was reported only in three JIA patients, and none of the children in the control group. One patient reported mild joint pain, the others had no symptoms of COVID-19 disease. There was no exacerbation of JIA symptoms in any of the SARS-CoV-2 positive children.

Contact with COVID-19 (without RT-PCR test confirmation) was reported by a few children: one in the JIA group (positively tested for SARS-CoV-2 IgG and IgA antibodies), five in the control group (three positive for IgG and IgA antibodies, one in IgG only, and in two both results were negative).

In the JIA group, the number of active joints (AJN) was significantly higher than in the SARS-CoV-2 IgA-positive patients. It should be noted that the mean values of disease activity, according to the Juvenile Arthritis Disease Activity Score of all 71 joints (JADAS 71) [6], were higher in both IgA and IgG positive patients, although these differences were not statistically significant (Table 3).

We found no difference in seropositivity depending on the biological treatment of disease-modifying antirheumatic drugs (DMARDs), methotrexate, and GC. Among patients treated with sulfasalazine, there was a statistically higher number of positive anti-SARS-CoV-2 IgA tests. Interestingly, patients treated with hydroxychloroquine were significantly more frequently IgG-seropositive. However, it should be noted that these groups were too small to draw unequivocal conclusions.

### 2.3. PD-1 Serum Concentration in JIA Patients

There were no statistically significant differences in PD-1 levels between patients with JIA and controls (Table 2).

Significantly higher PD-1 serum concentration was observed in both patients with JIA and the control group who were seropositive for SARS-CoV-2 IgA or IgG antibodies compared to those who were seronegative (Table 3).

The study also showed a positive correlation of PD-1 concentration with the ratio of SARS-CoV-2 IgA and IgG antibodies, both in the whole study group and in the group of JIA patients (Table 4).

In the JIA group, no relationship between the concentration of PD-1, the type of used drugs, or disease activity was found (Table 5).

### 2.4. Prevalence of Seropositivity in JIA Patients Depending on Earlier Vaccinations

The study did not show any association of seropositivity against SARS-CoV-2 with earlier flu vaccinations (as a representative of vaccination against viral infections) or pneumococci vaccinations (as a representative of vaccination against bacterial infections) (Table 6).

## 3. Discussion

Our study showed a higher prevalence of positive results of anti-SARS-CoV-2 antibodies in children with JIA compared to the control group (IgA 29.2% vs. 6.5%, IgG: 24.6% vs. 12.9%, respectively). Positive results of either IgA or IgG were observed in 33.8% of the JIA patients while in the control group only 12.9% of children were seropositive.

Our study was carried out from 1 October 2020 to 30 April 2021, during the second wave of the COVID-19 pandemic in Poland, with the number of cases from 7000 to 31,000 per day/39.3 million inhabitants [7]. That was before the introduction of COVID-19 vaccination for children under the age of 18, therefore none of the subjects were vaccinated.

In our previous study conducted during the first wave of the COVID-19 pandemic (from 1 June to 30 September 2020), with a confirmed number of cases from 362 to 1405 per day/39.3 million inhabitants, the prevalence of anti-SARS-CoV-2 antibodies in children with JIA was assessed [4,7]. At that time, lower seropositivity in children with JIA (12.9% positive in at least one of the antibodies IgA or IgG) and a slightly higher seropositivity in healthy children was noted [4].

Therefore, we believe that two factors should be further considered in this topic. First, the current state of the immune system of children with JIA (including therapy), as well as frequent contact with the healthcare facilities (including routine medical check-ups of patients with JIA every 4–8 weeks), should be considered. During the period covered by the study, there was a lockdown in Poland and school education was conducted online at home.

Despite the high incidence of COVID-19, contact with an infected person was reported only by four JIA patients (all of them tested positive for SARS-CoV-2 IgG and IgA antibodies), with three having a history of COVID-19 diagnoses confirmed by RT PCR test. No serious symptoms or exacerbations of JIA were observed in any patient diagnosed with COVID-19.

Our studies did not establish a clear relationship between IgA or IgG seropositivity with disease activity assessed by JADAS 71 or used treatment.

Similar studies in children with rheumatic diseases have been conducted in other countries [8,9,10]. In a study conducted in Turkey from March 2020 to August 2020 [8], exactly in the period of our first study [4], 149 children asymptomatic for COVID-19 (43 healthy children, 42 with JIA, and 64 with other rheumatic diseases) were included. The proportion positive for anti-SARS-CoV-2 IgA or IgG was 12.75% children, including 18.6% healthy children and 11.9% diagnosed with JIA, but those differences were not statistically significant. Both IgA and IgG positivity were not found to be related to age, underlying rheumatic diseases, and treatment. Disease activity was not assessed in this study [8].

In a study from New York conducted from May to November 2020, focused on 262 children with rheumatoid diseases (184 with JIA) with at least one immunosuppressive medication, 13% (14.7% with JIA) were SARS-CoV-2 IgG positive and 6.4% had symptoms suggestive of COVID-19 (fever, fatigue, and cough). No SARS-CoV-2 IgG positive children developed severe or critical COVID-19 or required hospitalization. Patients on medications such as TNF inhibitors, methotrexate, hydroxychloroquine, or corticosteroids did not exhibit higher rates of SARS-CoV-2 IgG positivity [9].

In a study from Germany, the clinical manifestations, course, and outcome of SARS-CoV-2 infection in 76 children with rheumatic diseases (44 with JIA) from April 2020 to February 2021 were analyzed. In 76% of patients under various immunosuppressive medications, SARS-CoV-2 infection was mild with good outcome in the majority of cases and did not have a relevant impact on rheumatic disease activity [10].

Both the abovementioned observations, as well as our own results, indicate a mild course of SARS-CoV-2 infection in children with JIA, regardless of disease activity and treatment. However, the constantly changing epidemiological situation and emerging new variants of the virus have an impact on the results.

Our research did not show any association between seropositivity SARS-CoV-2 antibodies with earlier flu vaccinations (as a representative of vaccination against viral infections) or pneumococci vaccinations (as a representative of vaccination against bacterial infections).

There were also no significant differences in PD-1 levels between patients with JIA and the control group. Significantly higher PD-1 serum concentration was observed in patients with JIA and control patients who were seropositive for anti-SARS-CoV-2 IgA or IgG antibodies compared to those who were seronegative. The study also showed a positive correlation between PD-1 concentration with the level (ratio) of SARS-CoV-2 IgA and IgG antibody, both in the whole study group, as well as in the group of JIA patients. In the JIA group, no relationship between the concentration of PD-1, the type of drug used, and disease activity was found.

PD-1 (or CD279) is a member of the CD28 superfamily, expressed on CD4+ and CD8+ T cells, NK cells, B cells, and activated monocytes. Binding to its ligands PD-L1 and PD-L2 on T cells, B cells, dendritic cells (DCs), and macrophages, PD-1 conducts its inhibitory function to regulate T cell activation, tolerance, and immune-mediated tissue damage [5].

The main function of the PD-1/PD-1L pathway is regulation of the T-cell function, especially during persistent antigenic stimulation, such as chronic viral infections, tumors, or stimulation by self-antigens. This pathway controls multiple tolerance checkpoints that prevent autoimmunity [2,11].

In chronic viral infections, PD-1 overexpression causes T-cell exhaustion and impairs the ability to destroy the infectious cells. According to current knowledge, the infection of the target cells by SARS-CoV-2 starts the activation of innate and acquired immune systems, including T cells and dendritic cells (DC). These immune cells secrete inflammatory cytokines such as IL-10 and INF-γ. IL-10 in turn increases the expression of PD-1 on the monocytes and DCs and suppresses the function and differentiation of CD4+ T-cells. The attachment between PD-L1 on monocytes and the PD-1 on the CD8+ T-cells inhibits the antiviral activity and could contribute to disease progression [2].

PD-1 and its ligand (PD-L1) mediate negative signals in autoimmune diseases, but the role in the pathogenesis of JIA is poorly understood. Several studies have demonstrated that the PD-1/PD-L1 pathway is involved in autoimmune diseases mediated by T cells such as autoimmune type 1 diabetes (DMI), rheumatoid arthritis (RA), and systemic lupus erythematosus (SLE). The study of Cai et al. assessed the PD-1/PD-L1 signaling in JIA patients and analyzed the association with disease activity and clinical manifestations. They found that patients with active systemic JIA had lower PD-1 expression as compared to healthy controls and patients with active polyarthritis and enthesitis-related arthritis (ERA). Additionally, they had lower PD-L1 expression on DC compared with healthy controls. Both PD-1 on CD4+ T cell and PD-L1 on DC were negatively correlated with JADAS-27 in systemic JIA (sJIA) patients [5]. Significantly lower PD-L1 expression on myeloid cells in sJIA was also reported compared with other febrile patients [12]. These studies suggest a possible role for the PD-1 pathway in JIA. Nevertheless, in our group of JIA patients, there were only two patients with sJIA, and both of them had inactive disease at the time of the study.

Petrelli et al. [13] showed an overrepresentation of PD-1+CD8+ T cells in the synovial fluid (SF) of JIA patients. Gene expression profiling and other tests identified PD-1+CD8+ T cells as metabolically active effectors, with no apparent proof of their exhaustion. The induction and expansion of CD8+ T cells with potential damaging properties in chronic arthritis were increased. According to the authors, these findings provide the basis for further research of PD-1-expressing CD8+ T cell targeting strategies in JIA [13].

Similarly, a study in RA adult patients demonstrated that the expression of PD-1 is upregulated on CD4+ T and CD8+ T cells in peripheral blood and synovial fluid and established a correlation between the expression of PD-1 on T cells and the disease activity of RA [14].

In our study, serum levels of PD-1 were higher in JIA patients with serological evidence of prior SARS-CoV-2 infection and no active infection, even though most likely 4-8 weeks or more had passed from the exposure to the virus.

Importantly, the concentration of PD-1 did not correlate with the erythrocyte sedimentation rate (ESR), C-reactive protein (CRP) levels, and age in the entire group and JIA, and did not correlate with disease activity in the JIA group. This would mean that following SARS-CoV-2 infection, increased PD-1 pathway activity persists for many weeks, regardless of whether the child is healthy or has JIA diagnosed and, in the case of JIA, regardless of whether the disease is active.

CD8 T-cell exhaustion related to increased expression of PD-1 could be associated with a good prognosis in multiple autoimmune diseases, but with poor outcomes in chronic infections. Potentially, increased expression of PD-1 might lead to CD8+ T cell exhaustion and reduced immunity not only against viral infections but also against tumors.

This is another reason to carefully monitor children with a history of COVD-19 and, above all, to protect against SARS-CoV-2 infection. We are currently conducting a similar study in JIA patients vaccinated against COVID-19 and hope to be able to communicate our results soon.

To our knowledge, no similar reports have been published so far in other rheumatic diseases in adults and children. However, the observation that the association of another viral infection, EBV, with the activity of the PD-1/PD-L1 axis in patients with SLE is interesting. PD-L1 expression is upregulated on SLE patient peripheral blood neutrophils but reduced on dendritic cells and monocytes. EBV preferentially infects B cells. One of the common EBV proteins is latent membrane protein 1 (LMP1). NF-κB activation induced by LMP1 generates various cytokines, including IFN, which plays a role in PD-L1 expression in infected cell lines. Possibly, these same signals regulate peripheral blood neutrophil PD-L1 over-expression in SLE. T cell PD-L1 expression is also induced by the EBV cytokine IL-27. Increased levels of PD-1 are expressed by activated B cells in SLE. Because PD-L1 antagonizes TCR and BCR cell signals, the presence of sPD-L1 in SLE may directly affect the activity of lymphocytes [15].

The limitation of our study was the inability to assess the time between exposure to COVID-19 and the outcome, due to the asymptomatic course of most cases.

We did not assess lymphocyte counts in relation to PD-1 concentration because in JIA patients it is associated with many factors, including disease activity and treatment.

We also see a need to evaluate the expression of PD-1 and PD-L1 molecules on immune cells.

Further research is required to check whether the relationships we have found are specific to JIA or whether they occur in other rheumatic diseases in children and adults.

In summary, the PD-1/PD-Ls pathway is a key regulator in T-cell activation and tolerance, and it plays crucial roles in autoimmunity, infectious immunity, and tumor immunity. Our study indicates that any of these pathways may be affected by a SARS-CoV-2 infection in both healthy and JIA patients. Nevertheless, we can only recommend careful observation at present.

## 4. Materials and Methods

### 4.1. Study Group

All subjects were diagnosed and treated in the Department of Paediatric Pulmonology and Rheumatology, Medical University of Lublin (Poland).

Sera were obtained from a total of 96 patients who were admitted to our department from 1 October 2020 to 30 April 2021, during the second wave of the COVID-19 pandemic in Poland. The study was conducted in the period prior to the introduction of COVID-19 vaccinations for the children at the age before 18, as such none of the subjects were vaccinated.

An antigen test for SARS-CoV-2 was performed on each subject on the day of admission, and a positive result was the exclusion criterion.

The JIA group (JIA diagnosed according to the International League of Associations for Rheumatology [ILAR] criteria [16]) included patients aged 2–18 years with oligoarthritis (*n* = 25), polyarthritis with positive rheumatoid factor (RF) (*n* = 5), polyarthritis with negative RF (*n* = 15), psoriatic arthritis (*n* = 2), enthesitis-related arthritis (*n* = 11), and systemic arthritis (*n* = 2). All patients diagnosed with JIA were treated with DMARDs for at least 3 months, alone or in combination with: conventional synthetic (methotrexate, sulfasalazine, hydroxychloroquine) or biologic (etanercept, adalimumab, tocilizumab). A total of 14 patients were additionally receiving systemic glucocorticoids (orally, more than to 2 weeks, regardless of the dose). There was no patient without treatment in the JIA group.

The control group included 31 healthy children of health workers. We excluded children taking medication affecting the immune system; reporting symptoms of infection in the last three months before the study; or patients with diagnosed chronic diseases, such as allergies, inflammatory, autoimmune, or oncological diseases.

Sera were obtained from a total of 96 patients (65 JIA patients and 32 controls) during routine laboratory tests. Inflammatory markers, including ESR and CRP, were examined during routine outpatient visits. Clinical data were extracted from the electronic medical record.

In patients with JIA, the disease activity was estimated with the use of the juvenile arthritis disease activity score 71 (JADAS 71) [6]. The JADAS 71 includes the following four measures: physician’s global assessment of disease activity (PhGA) and parent global assessment of well-being (PGA), measured on a 0–10 visual analogue scale (VAS) where 0 = no activity and 10 = maximum activity; erythrocyte sedimentation rate (ESR), normalized to a 0 to 10 scale; and a count of joints with active disease [6].

### 4.2. Detection of Anti-SARS-CoV-2 Antibodies

ELISA based tests for anti-SARS-CoV-2 IgA and IgG were from Euroimmun (Lubeck, Germany). These tests were used according to their manufacturer’s instructions. Results were calculated as the absorbance value of the sample divided by the absorbance value of the calibrators and expressed as extinction ratio. We utilized the manufacturer’s interpretation of the ratio with samples < 0.8 classified as no antibody present, 0.8–<1.1 indeterminate, and ≥1.1 containing antibodies. These ELISA tests check for antibodies against the S1 subunit/domain of the spike protein of SARS-CoV-2.

### 4.3. Detection of Serum PD-1

For assessment of serum PD-1 levels, an ELISA kit (Invitrogen, Waltham, MA, USA) was used according to the manufacturer’s instructions. To determine the concentration of circulating human PD-1 for each sample, the mean absorbance value and the corresponding concentration of human PD-1 were found. The assay detects both natural and recombinant human PD-1.

### 4.4. Compliance with Research Ethics Standards

All patients and parents or legal guardians were informed in detail in oral and written form about the course, aims, and scope of the conducted research. All patients over 16 years and parents or guardians signed an informed written consent to participate in the study. The study was carried out in compliance with the Declaration of Helsinki. The study design was approved by the Bioethics Committee at the Medical University of Lublin (KE-0254/236/2020).

### 4.5. Statistical Analyses

Results from measurable parameters are presented as the mean, median, minimum, as well as maximum values and standard deviation. Immeasurable parameters are presented as means of count and percentage. The normal distribution of variables was checked using the Shapiro–Wilk test. Student’s *t*-test and the Mann–Whitney U test were used for intergroup comparisons for normally and non-normally distributed data, respectively. Differences between more than two groups were analyzed with the Kruskal–Wallis test, ANOVA, and multiple comparisons of mean ranks (as post hoc analysis) with the Bonferroni correction. The associations between pairs of variables were assessed with Spearman’s rank correlation. Statistical significance was considered at *p* < 0.05. The statistical analysis was carried out using Statistica 13.3 software (StatSoft, Kraków, Poland).

## 5. Conclusions

JIA patients have a higher seroprevalence of anti-SARS-CoV-2 antibodies than healthy subjects. Humoral immune response to SARS-CoV-2 infection is connected with persistent activation of PD-1 expression both in patients with JIA and in healthy children. The clinical significance of the detected disorder requires further careful observation.

## Figures and Tables

**Table 1 jcm-11-04060-t001:** Baseline demographic and clinical characteristics of the patients.

Parameter	JIA (*n* = 65)	Control (*n* = 31)	*p* Value
	Median (Range)	Median (Range)	
Age (years)	11.0 (2.0–17.0)	10.0 (2.0–18.0)	0.065
ESR (mm/h)	13.0 (2.0–121)	8.0 (2.0–30.0)	0.14
CRP (mg/dL)	0.0 (0.0–7.5)	0.0 (0.0–1.2)	1.0
Anti-pneumococcal vaccination	10 (15.4%)	12 (38.7%)	0.018
Anti-influenza vaccination	7 (10.8%)	3 (9.7%)	1.0
Contact with COVID-19 case	3 (4.6%)	6 (19.4%)	0.054
Disease activity		ND	ND
JADAS 71	5.25 (0.0–38.0)		
Active joint number	1.0 (0.0–24.0)		
PGA	2.0 (0.0–7.0)		
PhGA	2.0 (0.0–8.0)		
Treatment		ND	ND
Conventional synthetic DMARDs	60 (92,3%)		
Methotrexate	48 (73.9%)		
Hydroxychloroquine	5 (7.7%)		
Sulfasalazine	13 (20.0%)		
GC	14 (21.5)		
Biological DMARDs	33 (50.8%)		
Adalimumab	18 (54.6%)		
Etanercept	11 (33.3%)		
Tocilizumab	4 (12.1%)		

JIA, juvenile idiopathic arthritis; CRP, C-reactive protein; ESR, erythrocyte sedimentation rate; JADAS 71, juvenile arthritis disease activity score 71; PhGA, physician global assessment of disease activity; PGA, parent/patient assessment of overall well-being; ND, not determined, DMARDs, disease-modifying antirheumatic drugs; GC systemic glucocorticoids (orally, more than to 2 weeks, regardless of the dose).

**Table 2 jcm-11-04060-t002:** Immunological characteristics of the patients.

Parameter	JIA (*n* = 65)	Control (*n* = 31)	*p* Value
	Median (Range)	Median (Range)	
PD-1 serum concentration (pg/mL)	17.5 (3.0–112–8)	16.43 (2.7–48.2)	0.42
IgA anti-SARS-CoV-2 (ratio)	0.86 (0.07–14.1)	0.24 (0.07–12.6)	<0.0001
Positive anti-SARS-CoV-2 IgA	19 (29.2%)	2 (6.5%)	0.016
IgG anti-SARS-CoV-2 (ratio)	0.16 (0.05–8.8)	0.11 (0.05–9.3)	0.0025
Positive anti SARS-CoV-2 IgG	16 (24.6%)	4 (12.9%)	0.28

PD-1-programmed death receptor 1, JIA—juvenile idiopathic arthritis.

**Table 3 jcm-11-04060-t003:** Parameters of JIA patients and control dependence on the result of anti-SARS-CoV-2 antibodies (seropositive versus seronegative).

Parameter	IgA Anti-SARS-CoV-2				IgG Anti-SARS-CoV-2			
MIZS	Positive (*n* = 19)	Negative (*n* = 46)	Z/RR (95% CI)	*p* Value	Positive (*n* = 16)	Negative (*n* = 49)	Z/RR (95% CI)	*p* Value
Age (years)	12.0 (3–17)	11.0 (3–17)	−0.28	0.78	12.0 (4–16)	11.0 (3–17)	−0.038	0.97
ESR (mm/h)	15.0 (2–121)	12.0 (0–81)	0.995	0.32	12.0 (2–61)	13.0 (2–121)	−0.18	0.86
CRP (mg/dL)	0.03 (0–6.75)	0.0 (0–7.5)	0.084	0.35	0.015 (0–2.13)	0.0 (0–7.5)	−0.13	0.89
PD-1 (pg/mL)	25.7 (7.2–112.8)	15.2 (3.0–87.0)	2.62	0.0089	75.9 (7.2–112.8)	15.2(3.0–57.3)	3.40	0.00067
JADAS 71	8.85 (0–38)	5.0 (0–33.1)	1.12	0.26	11.0 (0–38)	5.0 (0–33.1)	1.60	0.11
AJN	1.0 (0–24)	1.0 (0–13)	0.64	0.52	4.5 (0–24)	1.0 (0–13)	1.97	0.049
PGA	2.0 (0–7)	2.0 (0–7)	0.30	0.77	3.0 (0–7)	2.0 (0–7)	1.22	0.22
PhGA	2.0 (0–7)	1.5 (0–8)	0.64	0.52	2.5 (0–8)	1.0 (0–7)	1.07	0.28
Therapy								
bDMARD	9 (47.4%)	24 (52.2%)	0.9 (0.5–1.6)	0.79	7 (43.8%)	26 (53.1%)	0.8 (0.4–1.5)	0.57
Methotrexate	13(68.4%)	35 (76.1%)	0.9 (0.6–1.3)	0.55	12 (75.0%)	36 (73.5%)	1.0 (0.7–1.4)	1.0
Hydroxy-chloroquine	2 (10.5%)	3 (6.5%)	1.6 (0.3–8.9)	0.62	4 (25.0%)	1 (2.0%)	12.3 (1.5–101.8)	0.011
Sulfasalazine	8 (42.2%)	5 (10.9%)	3.9 (1.5–10.3)	0.014	5 (31.3%)	8 (16.3%)	1.9 (0.7–5.0)	0.28
GC	6 (31.6%)	8 (17.4%)	1.8 (0.7–4.5)	0.32	3 (18.8%)	11 (22.5%)	0.8 (0.3–2.6)	1.0
**Control**	**Positive (*n* = 2)**	**Negative (*n* = 29)**	**Z/RR (95% CI)**	***p* Value**	**Positive (*n* = 4)**	**Negative (*n* = 27)**	**Z/RR (95% CI)**	***p* Value**
Age (years)	5.5 (2–9)	9.0 (1–18)	0.68	0.49	3.0 (2–9)	9.0 (1–18)	1.2	0.22
ESR (mm/h)	12.5 (5–20)	8.0 (2–30)	−0.44	0.66	11.5 (4–20)	8.0 (2–30)	−0.47	0.64
CRP (mg/dL)	0 (0)	0 (0–1.2)	1.00	0.31	0.0 (0–0.2)	0.0 (0–1.2)	0.65	0.52
PD-1 (pg/mL)	42.4 (36.6–48.2)	16.1 (2.7–45.3)	−2.13	0.033	42.8 (36.6–48.2)	13.3 (2.7–30.1)	−3.15	0.0016

JIA, juvenile idiopathic arthritis; AJN, active joint number; CRP, C-reactive protein; ESR, erythrocyte sedimentation rate; JADAS 71, juvenile arthritis disease activity score 71; PhGA, physician global assessment of disease activity; PGA, parent/patient assessment of overall well-being; bDMARDs, biological disease-modifying antirheumatic drugs; GC systemic glucocorticoids (orally, more than to 2 weeks, regardless of the dose), PD-1—programmed death receptor 1.

**Table 4 jcm-11-04060-t004:** Correlations (Spearman) for all subjects (*n* = 96).

Group	All Subjects (96)			JIA Patients (65)		
Correlation	Spearman R	t(N2)	*p* Value	Spearman R	t(N2)	*p* Value
PD-1 [pg/mL] and IgA anti-SARS-CoV-2 ratio	0.25	2.50	0.014	0.23	1.95	0.049
PD-1 [pg/mL] and IgG anti-SARS-CoV-2 ratio	0.32	3.26	0.0015	0.33	2.78	0.0073
PD-1 [pg/mL] and ESR	−0.027	–0.26	0.79	–0.038	–0.301	0.76
PD-1 [pg/mL] and CRP	0.058	0.560	0.58	0.12	0.96	0.34
PD-1 [pg/mL] and age	–0.045	–0.43	0.67	–0.063	–0.50	0.62
PD-1 [pg/mL] and JADAS 71	ND	ND	ND	–0.060	–0.47	0.64

JIA, juvenile idiopathic arthritis; ND, not determined, PD-1—programmed death receptor 1.

**Table 5 jcm-11-04060-t005:** Comparison of PD-1 levels depending on the therapy and disease activity (JADAS 71δ1 versus JADAS 71 > 1).

Parameter	PD-1	*p*-Value
	Median (Range)	
Therapy		
Patients with bDMARDs (*n* = 33)	16.7 (4.2–112.8)	0.62
Without (*n* = 32)	18.4 (3.0–106.4)	
Control (*n* = 31)	16.4 (2.7–48.2)	
Patients with Methotrexate (*n* = 48)	19.0 (3.0–112.8)	0.60
Without (*n* = 17)	15.1 (5.4–87.0)	
Control (*n* = 31)	16.4 (2.7–48.2)	
Patients with Hydroxychloroquine (*n* = 5)	80.2 (35.3–104.8)	0.0062
Without (*n* = 60)	16.4 (3.0–-112.8)	
Control (*n* = 31)	16.4 (2.7–48.2)	
Patients with Sulfasalazine (*n* = 13)	17.5 (7.2–106.4)	0.69
Without (*n* = 52)	17.7 (3.0–112.8)	
Control (*n* = 31)	16.4 (2.7–48.2)	
Patients with GC (*n* = 16)	25.3 (8.1–106.4)	0.066
Without (*n* = 49)	15.2 (3.0–112.8)	
Control (*n* = 31)	16.4 (2.7–48.2)	
Disease activity		
Patients with JADAS 71 > 1 (*n* = 41)	16.6 (3.0–106.4)	0.13
With JADAS 71 ≤ 1 (*n* = 24)	25.1 (5.3–112.8)	
Control (*n* = 31)	16.4 (2.7–48.2)	

PD-1—programmed death receptor 1, bDMARDs, biological disease-modifying antirheumatic drugs; JADAS 71, juvenile arthritis disease activity score 71.

**Table 6 jcm-11-04060-t006:** The prevalence of seropositivity in JIA patients depending on earlier vaccinations: anti-pneumococcal and anti-influenza.

Parameter	IgA Anti-SARS-CoV-2				IgG Anti-SARS-CoV-2			
JIA	Positive (*n* = 19)	Negative (*n* = 46)	RR (95% CI)	*p* Value	Positive (*n* = 16)	Negative (*n* = 49)	RR (95% CI)	*p* Value
Anti-pneumococcal vaccination	3 (15.8%)	7 (15.2%)	1.0 (0.3–3.6)	1.0	2 (12.5%)	8 (16.3%)	0.8 (0.2–3.2)	1.0
Anti-influenza vaccination	1 (5.3%)	6 (13.0%)	0.4 (0.05–3.1)	0.66	0 (0)	7 (14.3%)	-	0.18

JIA, juvenile idiopathic arthritis.

## Data Availability

The datasets used and/or analyzed during the current study are available from the corresponding author on reasonable request.

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
