# Peer review of "Programmed Cell Death Protein-1 Upregulation in Response to SARS-CoV-2 in Juvenile Idiopathic Arthritis: A Case-Control Study"

_jcm, 2022, doi:10.3390/jcm11144060_

Round 1

Reviewer 1 Report

The paper reports on PDL1 expression levels in JIA SARS+ children in comparison to healthy controls. The cohort is well described; statistical tests as well. Comments are given below.

1) I would strongly suggest including measurements in another autoimmune disease, child and/or adult onset, RA and SLE could be options. This would show if the finding of PD1 connection to SARS is JIA specific or not.

2) another method needs to be included to support results of ELISA - Western blot, PCR, or other.

3) Discussion part, consider adding how your findings could benefit treatment/monitoring of JIA SARS+

Author Response

We carefully considered all the comments from the reviewers.

1) I would strongly suggest including measurements in another autoimmune disease, child and/or adult-onset, RA and SLE could be options. This would show if the finding of PD1 connection to SARS is JIA specific or not.

We thank you for this very valuable remark. We plan to extend our research to other rheumatic diseases in children. However, to the best of our knowledge, no similar reports have been published so far regarding other rheumatic diseases in adults and children.

We have added a comment in the discussion.

Similarly, study in RA adult patients demonstrated that the expression of PD-1 is upregulated on CD4+ T and CD8+ T cells in peripheral blood and synovial fluid and established a correlation between the expression of PD-1 on T cells and the disease activity of RA this [15]."

"To our knowledge, no similar reports have been published so far in other rheumatic diseases in adults and children. However, the observation the association of another viral infection, EBV, with the activity of the PD-1 / PD-L1 axis in patients with SLE is interesting. PD-L1 expression is upregulated on SLE patient peripheral blood neutrophils but reduced on dendritic cells and monocytes. EBV preferentially infects B cells. One of common EBV proteins is latent membrane protein 1 (LMP1). NF-κB activation induced by LMP1 generates various cytokines, including IFN that plays a role in PD-L1 expression in infected cell lines. Possibly, these same signals regulate peripheral blood neutrophil PD-L1 over-expression in SLE. T cell PD-L1 expression is also induced by the EBV cytokine IL-27. Increased levels of PD-1 are expressed by activated B cells in SLE. Because PD-L1 antagonizes TCR and BCR cell signals the presence of sPD-L1 in SLE may directly affect the activity of lymphocytes [16]."

"Further research is required to check whether the relationships we have found are specific to JIA or whether they occur in other rheumatic diseases in children and adults."

2) Another method needs to be included to support results of ELISA - Western blot, PCR, or other.

Thank you for your comment. Unfortunately, due to some limitations regarding the collection of material from children, which was primarily used to perform laboratory tests necessary to control the underlying disease, we were unable to use other methods of assessing PD-1 receptor expression. Nevertheless, in the future, we aim to evaluate PD-1 receptor expression on individual T cell subpopulations in the same patients.

3) Discussion part, consider adding how your findings could benefit treatment/monitoring of JIA SARS+

We have added a comment in the discussion.

"In summary, the PD-1/PD-Ls pathway is a key regulator in T-cell activation and tolerance, and it plays crucial roles in autoimmunity, infectious immunity, and tumor immunity. Our study indicates that any of these pathways may be affected by a SARS-COV-2 infection in both healthy and JIA patients. Nevertheless, we can only recommend careful observation at present."

Reviewer 2 Report

The topic of this manuscript is interesting.

(1) The English writing need extensive polishing. 

(2) When the patients were diagnosed with Juvenile Arthritis Disease, before pandemic of COVID-19 or during COVID-19? Does COVID-19 accelerated the onset of Juvenile Arthritis Disease?

(3)  Why both parametric (t-test) and non-parametric test (Mann–Whitney U test) were used in the same study? Does the data distributed normally? 

(4) The incidence rates of seropositive  SARS-CoV-2 IgA or IgG antibodies were high. However, did the patients suffer from COVID-19 symptoms or had confirmed medical record of COVID-19? 

Author Response

We carefully considered all the comments from the reviewers.

1) The English writing need extensive polishing. 

We apologize for the mistakes. The manuscript has been corrected accordingly.

2) When the patients were diagnosed with Juvenile Arthritis Disease, before pandemic of COVID-19 or during COVID-19? Does COVID-19 accelerated the onset of Juvenile Arthritis Disease?

We did not conduct a JIA incidence analysis before and during the pandemic, although it is a very interesting topic. Especially that our observations as practitioners indicate less incidence. This is a multi-factorial topic, including less access to a doctor (telepaths), less activity of children (home schooling) and a change in exposure to many environmental factors, including common infections.

We have completed the description as below.

“All JIA patients were diagnosed according to the ILAR 2001 classification [5] prior to study enrollment and were treated for at least 6 weeks with at least one conventional or biological disease-modifying antirheumatic drugs (DMARDs).”

(3)  Why both parametric (t-test) and non-parametric test (Mann–Whitney U test) were used in the same study? Does the data distributed normally? 

Thank you for this comment. We added more detailed information regarding the statistical analysis. Normally distributed data were calculated with parametric tests (Student t-test, ANOVA). Some variables do not have normal distribution so the non-parametric tests were used (Mann-Whitney U test, Kruskal-Wallis test).

“The normal distribution of variables was checked using the Shapiro–Wilk test. The Student t-test and the Mann–Whitney U test were used for intergroup comparisons for normally and non-normally distributed data, respectively.”

(4) The incidence rates of seropositive  SARS-CoV-2 IgA or IgG antibodies were high. However, did the patients suffer from COVID-19 symptoms or had confirmed medical record of COVID-19? 

We described the clinical characteristics of patients

“COVID-19 confirmed by SARS-CoV-2 RT-PCR test (4-8 weeks before our tests) was reported only in 3 JIA patients, and none of the children in the control group. One patient reported mild joint pain, the others had no symptoms of COVID-19 disease. There was no exacerbation of JIA symptoms in any of the SARS-CoV-2 positive children.

Contact with COVID-19 (without RT-PCR test confirmation) was reported by a few children: respectively 1 in the JIA group (positively tested for SARS-COV-2 IgG and IgA antibodies), 5 in the control group (3 positive for IgG and IgA antibodies, 1 in IgG only, and in 2 both results were negative)."

Round 2

Reviewer 2 Report

The manuscript has been improved and it  appears to be acceptable.